# Lenvatinib as First-Line Treatment for Unresectable Hepatocellular Carcinoma: A Systematic Review and Meta-Analysis

**DOI:** 10.3390/cancers14225525

**Published:** 2022-11-10

**Authors:** Shijie Wang, Yiting Wang, Jiangtao Yu, Huaxing Wu, Yanming Zhou

**Affiliations:** 1Department of Oncological Surgery, First Affiliated Hospital of Xiamen University, Xiamen 361000, China; 2Department of General Surgery, Jinling Hospital, Medical School of Nanjing University, Nanjing 210000, China; 3Department of Stomatology, People’s Hospital of Zhengzhou, People’s Hospital of Henan University of Chinese Medicine, Zhengzhou 450000, China; 4Department of General Surgery, People’s Hospital of Zhengzhou, People’s Hospital of Henan University of Chinese Medicine, Zhengzhou 450000, China

**Keywords:** lenvatinib, sorafenib, hepatocellular carcinoma, review, meta-analysis

## Abstract

**Simple Summary:**

Approximately 80% hepatocellular carcinoma (HCC) patients are in intermediate or advanced stages at diagnosis and have lost the chance of curative surgery, resulting in poor prognosis. Lenvatinib was approved in 2018 as a first-line treatment for unresectable HCC. The aim of this study was to evaluate the efficacy and safety of lenvatinib as a first-line treatment for unresectable HCC by reviewing the currently available data and comparing lenvatinib with sorafenib. An overview of 24 studies indicates that lenvatinib can provide better tumor responses and survival benefits as compared with sorafenib for unresectable HCC patients, with a comparable incidence of adverse events.

**Abstract:**

Lenvatinib was approved in 2018 as a first-line treatment for patients with unresectable hepatocellular carcinoma (HCC). This systematic review and meta-analysis aimed to provide the most updated evidence about the efficacy and safety of lenvatinib as a first-line treatment for unresectable HCC. An electronic search of the PubMed database, Web of Science, Embase, and Cochrane Library was undertaken to identify all relevant studies up to May 2022. The pooled effect sizes were calculated based on the random-effects model. One phase III randomized controlled trial and 23 retrospective studies of 2438 patients were eligible for analysis. For patients treated with lenvatinib as first-line treatment, the pooled median overall survival (OS), median progression-free survival (PFS), 1-year OS rate, 1-year PFS rate, objective response rate (ORR), and disease control rate (DCR) were 11.36 months, 6.68 months, 56.0%, 27.0%, 36.0% and 75.0%, respectively. Lenvatinib showed a significantly superior efficacy compared with sorafenib (HR for OS, 0.85 and HR for PFS, 0.72; OR for ORR, 4.25 and OR for DCR, 2.23). The current study demonstrates that lenvatinib can provide better tumor responses and survival benefits than sorafenib as a first-line treatment for unresectable HCC, with a comparable incidence of adverse events.

## 1. Introduction

Liver cancer is the 6th most common malignant tumor and the 4th leading cause of cancer-related death worldwide, with 841,000 new cases diagnosed per year [1]. Among primary liver cancers, hepatocellular carcinoma (HCC) represents the major histological subtype, accounting for 70–85% of all liver cancer cases [2]. Surgical resection and liver transplantation are effective treatments for patients with early stage HCC. However, due to the insidious onset, approximately 80% HCC patients are already in intermediate or advanced stages at diagnosis and have lost the chance of curative surgery, resulting in poor prognosis [3,4]. Based on the results from the SHARP and Asia-Pacific trials [5,6], sorafenib became the first approved systemic therapy for unresectable HCC in 2007. Lenvatinib, an oral multi-kinase inhibitor of vascular endothelial growth factor receptor (VEGFR) 1-3, platelet-derived growth factor receptor (PDGFR) α, fibroblast growth factor receptor (FGFR) 1-4, and RET and KIT proto-oncogenes, has previously been used in the treatment of papillary differentiated thyroid carcinoma and advanced endometrial cancer [7]. In 2018, supported by a phase III clinical trial, lenvatinib was also approved as a first-line treatment for unresectable HCC [8]. Since then, many researchers have assessed the treatment outcomes of lenvatinib for these patients. However, as most data are based on small single-center series, it is difficult to reach a consensus. In addition, the results of comparative studies of lenvatinib versus sorafenib are conflicting [9].

The aim of this study was to evaluate the efficacy and safety of lenvatinib as a first-line treatment for unresectable HCC by reviewing the currently available data, and comparing lenvatinib with sorafenib.

## 2. Materials and Methods

Systematic review and meta-analysis were conducted in accordance with the 2009 Preferred Reporting Items for Systematic Reviews and Meta-Analyses (PRISMA) Statement [10]. The protocol was registered with the International Prospective Register of Systematic Reviews (PROSPERO registration number: CRD42022356713). Meta-analysis of the present study strictly followed the PRISMA checklist for study search, screening, data extraction and analysis.

### 2.1. Search Strategy

An electronic search of PubMed database, Web of Science, Embase, and Cochrane Library was undertaken to identify all relevant studies published in English up to May 2022, using the following terms: hepatocellular carcinoma, HCC, liver cancer, lenvatinib and molecular-targeted agents. For additional citations, the reference list of retrieved articles was reviewed.

### 2.2. Study Selection

The literature search was conducted by two researchers (S.J.W and Y.T.W) independently. The inclusion criteria were as follows: (1) all study types except for case reports; (2) unresectable HCC patients receiving first-line treatment with lenvatinib as the study participants whose information could be extracted from a subgroup; and (3) studies that clearly reported the characteristics of the participants and post-treatment clinical responses.

### 2.3. Data Extraction

Data were extracted independently by two independent investigators (S.J.W and Y.T.W); any disagreement was resolved through discussion, or through further consultation with a third investigator (Y.M.Z). The first author, year of publication, country, study design, patient details, number of patients, overall survival (OS), progression-free survival (PFS), objective response rate (ORR), disease control rate (DCR), and adverse events (AEs) were extracted. For studies that did not directly indicate OS or PFS, information was derived from the reported survival plots. The hazard ratios (HRs) for OS and PFS with 95% confidence intervals (CIs) were extracted from each controlled study, or from the Kaplan–Meier curves in accordance with the protocol from Tierney et al. [11], if not available. For propensity score matched studies, only data after matching were considered.

### 2.4. Quality Assessment

The quality of the included studies was assessed using the Newcastle–Ottawa Scale (NOS) for cohort studies [12] or the Jadad Scale for randomized trials [13]. The studies with scores greater than or equal to 6 using NOS or that with scores greater than 2 using Jadad Scale were considered high quality.

### 2.5. Statistical Analysis

All statistical analyses were performed using Stata statistical software version 14.0. Survival data, including OS and PFS, are presented with HRs and 95% CI. Odds ratios (ORs) and 95% CIs were calculated for the analysis of tumor response and AEs. Meta-analyses of data were estimated with a random effects model. The I^2^ statistic was used to assess heterogeneity of the studies, with a value more than 50% defined as significant.

## 3. Results

### 3.1. Search Results

A total of 2132 articles from the databases and 36 articles from the reference list were identified by the initial search strategy, from which 1449 articles were excluded due to duplication. After screening the titles and abstracts, 464 unrelated studies were excluded, and by reading the full text, an additional 230 articles were excluded. One article was excluded because the study population had been included in a previous study [8]. Eventually, 24 studies involving patients with unresectable HCC were included in this analysis. A flowchart depicting the study selection is shown in Figure 1.

#### Patient Characteristics

Of the 24 studies included, one was a phase III randomized controlled trial (RCT) [8], and the other 23 were retrospective studies [8,9,14,15,16,17,18,19,20,21,22,23,24,25,26,27,28,29,30,31,32,33,34,35], involving 2438 patients with unresectable HCC who received first-line treatment with lenvatinib. The initial dose of oral lenvatinib was 12 mg QD for patients with a body weight of ≥60 kg, or 8 mg QD for those with a body weight of <60 kg. Of the 23 studies that assessed the tumor response, nineteen studies [9,14,15,16,17,18,19,20,22,23,24,25,27,28,31,32,33,34,35] applied the mRECIST, one [32] applied the RECIST 1.1, one [21] applied the RECIST 1.1 and mRECIST, and two [26,30] were unknown. Details of all studies and the characteristics of the patients are presented in Table 1.

### 3.2. Therapeutic Efficacy Assessment

#### 3.2.1. Overall Survival and Progression-Free Survival

The median OS, median PFS, 1-year OS rate and 1-year PFS rate were analyzed, respectively. Data were available on median OS in 10 studies involving 1120 patients, ranging from 4.2 to 15.2 months. The meta-analysis showed that the pooled median OS was 11.36 months (95%CI = 8.22–14.50) (heterogeneity analysis: I^2^ = 95.4%) (Figure 2A). Data were available on median PFS in 12 studies involving 1326 patients, ranging from 2.8 to 16.0 months. The meta-analysis also showed that the pooled median PFS was 6.68 months (95%CI = 5.38–7.98) (heterogeneity analysis: I^2^ = 93.7%) (Figure 2B).

The 1-year OS rate was available in 17 studies involving 1751 patients, ranging from 0% to 80.3%. The meta-analysis showed that the pooled 1-year OS rate was 56.0% (95%CI = 35.0–76.0%) (heterogeneity analysis: I^2^ = 99.4%) (Figure 3A). The 1-year PFS rate was available in 15 studies involving 1692 patients, ranging from 0% to 64.9%. The meta-analysis showed that the pooled 1-year PFS rate was 27.0% (17.0–36.0%) (heterogeneity analysis: I^2^ = 98.5%) (Figure 3B).

The I^2^ statistic indicated that data were heterogeneous in our analyses and therefore these summary measures should be interpreted with appropriate caution. Due to the limited studies in Western countries, subgroup analysis was not possible, and our results showed no significant decrease in heterogeneity after deleting the limited Western and multicenter studies (Appendix A).

#### 3.2.2. Objective Response Rate and Disease Control Rate

The ORR was available in 23 studies involving 2394 patients, ranging from 7.1% to 73.3%. The meta-analysis showed that the pooled ORR was 36.0% (95%CI = 29.0–44.0%) (heterogeneity analysis: I^2^ = 94.1%) (Figure 4A). The DCR was available in 21 studies involving 1856 patients, ranging from 38.2% to 100.0%. The meta-analysis showed that the pooled DCR was 75.0% (68.0–83.0%) (heterogeneity analysis: I^2^ = 97.0%) (Figure 4B). Heterogeneity was not substantially reduced after removing the limited Western and multicenter studies (Appendix A).

#### 3.2.3. Adverse Events

The AEs of unresectable HCC patients who received first-line treatment with lenvatinib are shown in Table 2. The most common AEs were hypertension (36.8%), followed by decreased appetite (36.4%), fatigue (34.5%), palmar–plantar erythrodysaesthesia syndrome (PPES) (29.4%), decreased weight (29.1%), proteinuria (25.6%), diarrhea (24.6%), and hypothyroidism (22.5%). Other less common AEs included elevated aspartate aminotransferase (AST) (18.0%), dysphonia (17.7%), vomiting (12.1%), nausea (16.4%), constipation (15.0%), abdominal pain (16.9%), decreased platelet count (18.2%), and bilirubin elevation (16.6%) (Table 2).

Not all studies reported severe forms of (grade 3–4) AEs. According to the valid data, the most common severe AEs were hypertension (12.5%), followed by proteinuria (6.02%), fatigue (5.9%), decreased appetite (5.8%), and decreased weight (5.6%) (Table 2).

### 3.3. Lenvatinib Versus Sorafenib

#### 3.3.1. Overall Survival

A total of 11 studies compared the OS of lenvatinib and sorafenib as first-line treatment for unresectable HCC patients. Of them, two studies showed that lenvatinib prolonged OS compared with sorafenib, and nine studies showed no significant difference. The meta-analysis showed that the pooled HR was 0.85 (95%CI = 0.73–0.99) (heterogeneity analysis: I^2^ = 41.1%), and lenvatinib mediated significantly better OS than Sorafenib (Figure 5A).

#### 3.3.2. Progression-Free Survival

A total of eight studies compared PFS between lenvatinib and sorafenib as first-line treatment for unresectable HCC patients. Of them, five studies showed that lenvatinib prolonged PFS compared with sorafenib, and three studies showed no significant difference. Meanwhile, the meta-analysis showed that the pooled HR was 0.72 (95%CI = 0.61–0.84) (heterogeneity analysis: I^2^ = 47.2%), and lenvatinib mediated significantly better OS than Sorafenib did (Figure 5B).

#### 3.3.3. Objective Response Rate

A total of 10 studies reported ORR of lenvatinib and sorafenib as first-line treatment for unresectable HCC patients. Of them, eight studies showed that lenvatinib was associated with a higher ORR as compared with sorafenib, and two studies showed no significant difference. The meta-analysis showed that the pooled OR was 4.25 (95%CI = 2.78–6.48) (heterogeneity analysis: I^2^ = 47.8%), and lenvatinib reported a higher ORR than sorafenib (Figure 6A).

#### 3.3.4. Disease Control Rate

A total of ten studies reported DCR of lenvatinib and sorafenib as first-line treatment for unresectable HCC patients. Of them, six studies showed a higher DCR compared with sorafenib, and four studies showed no significant difference. The meta-analysis showed that the pooled OR was 2.23 (95%CI = 1.70–2.93) (heterogeneity analysis: I^2^ = 39.0%), and lenvatinib reported a higher DCR than sorafenib (Figure 6B).

#### 3.3.5. Adverse Events

AEs between lenvatinib and sorafenib were compared based on the total number of AEs and severe (grade 3–4) AEs. The meta-analysis showed that there was no significant difference between the lenvatinib and sorafenib for all-grade (OR = 0.74, 95%CI: 0.20–2.79) (Figure 7A) and severe AEs (OR = 1.15, 95%CI: 0.88–1.50) (Figure 7B).

Subgroup analysis was performed on the 10 most reported AEs of lenvatinib and sorafenib in the original articles (Appendix A). The meta-analysis showed that in comparison with sorafenib, lenvatinib did not significantly increase the incidence of AEs including fatigue (OR = 1.54, 95%CI: 0.98–2.40) and elevated AST (OR = 0.78, 95%CI: 0.57–1.07). In contrast, lenvatinib significantly reduced the incidence of PPES (OR = 0.49, 95%CI: 0.26–0.95), diarrhea (OR = 0.69, 95%CI: 0.50–0.96) and rash (OR = 0.51, 95%CI: 0.28–0.93). However, for decreased appetite (OR = 1.97, 95%CI: 1.18–3.29), decreased platelet count (OR = 1.66, 95%CI: 1.19–2.32), hypertension (OR = 2.91, 95%CI: 1.86–4.54), hypothyroidism (OR = 12.05, 95%CI: 6.32–22.98) and proteinuria (OR = 2.82, 95%CI: 2.06–3.84), the incidence was significantly higher in the lenvatinib group.

### 3.4. Subgroup Analysis

Subgroup analyses were performed to explore the treatment effect of lenvatinib according to the eligibility criteria of the study population and the study background.

#### 3.4.1. Therapeutic Efficacy Assessment According to the Reflect Criteria

##### Overall and Progression-Free Survival

Eight of the ten studies with median OS did not meet the REFLECT criteria, and the meta-analysis showed that the pooled median OS was 10.80 months (95%CI = 7.17–14.43) (heterogeneity analysis: I^2^ = 95.6%) (Appendix A). Nine of the twelve studies with median PFS did not meet the REFLECT criteria, and the meta-analysis showed that the pooled median PFS was 6.44 months (95%CI = 5.01–7.86) (heterogeneity analysis: I^2^ = 94.5%) (Appendix A).

Fourteen of the seventeen studies with a 1-year OS rate did not meet the REFLECT criteria, and the meta-analysis showed that the pooled 1-year OS rate was 54.0% (95%CI = 32.0–77.0%) (heterogeneity analysis: I^2^ = 99.3%) (Appendix A); no significant difference was observed when compared with the overall population (*p* = 0.776). Twelve of the fifteen studies with a 1-year PFS rate did not meet the REFLECT criteria, and the meta-analysis showed that the pooled 1-year PFS rate was 28.0% (15.0–42.0%) (heterogeneity analysis: I^2^ = 97.6%) (Appendix A); no significant difference was observed when compared with the overall population (*p* = 0.874).

##### Objective Response Rate and Disease Control Rate

Eighteen of the twenty-three studies with ORR did not meet the REFLECT criteria, and the meta-analysis showed that the pooled ORR was 36.0% (26.0–45.0%) (heterogeneity analysis: I^2^ = 95.0%) (Appendix A); no significant difference was observed when compared with the overall population (*p* = 1.000). Eighteen of the twenty-three studies with DCR did not meet the REFLECT criteria, and the meta-analysis showed that the pooled DCR was 74.0% (65.0–83.0%) (heterogeneity analysis: I^2^ = 96.8%) (Appendix A); no significant difference was observed when compared with the overall analysis population (*p* = 0.871).

#### 3.4.2. Therapeutic Efficacy Assessment According to the Study Background

To make the included studies mimic the clinical settings as much as possible, nine studies representing only a specific population with different clinical settings were removed.

##### Overall Survival and Progression-Free Survival

Five of the ten studies with median OS were included, and the meta-analysis showed that the pooled median OS was 12.31 months (95%CI = 9.50–15.13) (heterogeneity analysis: I^2^ = 74.3%) (Appendix A). Six of the twelve studies with median PFS were included, and the meta-analysis showed that the pooled median PFS was 6.31 months (95%CI = 5.30–7.32) (heterogeneity analysis: I^2^ = 68.7%) (Appendix A).

Nine of the seventeen studies with a 1-year OS rate were included, and the meta-analysis showed that the pooled 1-year OS rate was 61.0% (95%CI = 57.0–65.0%) (heterogeneity analysis: I^2^ = 48.4%) (Appendix A). Eight of the fifteen studies with a 1-year PFS rate were included, and the meta-analysis showed that the pooled 1-year PFS rate was 29.0% (13.0–45.0%) (heterogeneity analysis: I^2^ = 98.3%) (Appendix A).

##### Objective Response Rate and Disease Control Rate

Fourteen of the twenty-three studies with ORR were included, and the meta-analysis showed that the pooled ORR was 35.0% (26.0–44.0%) (heterogeneity analysis: I^2^ = 95.1%) (Appendix A). Thirteen of the twenty-three studies with DCR were included, and the meta-analysis showed that the pooled DCR was 76.0% (70.0–82.0%) (heterogeneity analysis: I^2^ = 87.3%) (Appendix A).

## 4. Discussion

As many HCC patients have lost the chance of surgery at the time of diagnosis, it is primarily important to find an optimal systemic treatment for them. Currently, sorafenib and lenvatinib are the only approved first-line systemic agents for unresectable HCC. As lenvatinib was approved as the first-line treatment for unresectable HCC only a few years ago, reports about its therapeutic efficacy are inconsistent. It is therefore necessary to conduct a meta-analysis to assess its efficacy and safety. To the best of our knowledge, this is the first and largest single-arm meta-analysis to evaluate the efficacy and safety of lenvatinib versus sorafenib as a first-line treatment for unresectable HCC patients.

Antitumor efficacy is the most predominant cornerstone in evaluating lenvatinib. Our study showed that the pooled median OS was 11.36 months, which is worse than the result of the previous phase III clinical trial [8]. The possible reason is due to the presence of patients with Child–Pugh class C in some of our included studies [9,21,33]. When we excluded these studies for subgroup analysis, the result showed that the pooled median OS was 12.90 months (Appendix A). This obvious change also indirectly suggests that liver function is an important factor affecting the anti-tumor efficacy of lenvatinib. Regarding the survival benefit of lenvatinib in patients with hepatic decompensation (Child–Pugh class B and C), Park et al. reported a worse median OS (1.5 months) in the untreated group versus the lenvatinib group (4.2 months), which still could not prove the survival benefit of lenvatinib due to patient selection bias [9]. As with sorafenib, the first approved systemic agent for advanced HCC, lenvatinib is also recommended with caution in patients with hepatic decompensation due to the unsatisfactory survival benefits [36]. There is still some way to go before lenvatinib can be used as a first-line treatment for unresectable HCC patients with hepatic decompensation.

In addition to OS, PFS, ORR and DCR were also analyzed in this study. The results showed that the pooled median PFS, ORR and DCR were 6.68 months, 35.0% and 75.0%, respectively. Interestingly, the best median OS, median PFS, ORR, and DCR were obtained from the same study reported by Kudo et al. in 2019 [23]. The most likely reason for this favorable antitumor effect is that no patient included in their study had macroscopic vascular invasion and extrahepatic spread, and all had a Child–Pugh A liver function and Eastern Cooperative Oncology Group performance status of 0. These factors have proved to be favorable predictors of survival in patients with HCC, which are also important components of the REFLECT criteria [8,37]. However, not all patients could meet the REFLECT criteria in real-world settings, so we performed a subgroup analysis to evaluate the efficacy of lenvatinib in these patients. The results showed that the mOS and mPFS of patients who did not meet the REFLECT criteria were only slightly shorter than those in the overall population cohort, and there was no significant difference in tumor response. This proves that lenvatinib can still provide a favorable efficacy, even for patients who do not meet the REFLECT criteria. A similar result was also demonstrated in patients with advanced HCC in the study of Welland et al. [38], despite the fact that some surgically treated patients were included. When lenvatinib was compared with sorafenib, unlike the previous phase III clinical trial that showed no significant difference in OS, our meta-analysis showed that lenvatinib could significantly improve OS and PFS and offer better ORR and DCR. In addition, lenvatinib is significantly superior to sorafenib for patients with HBV infection in terms of OS (14.9 vs. 9.9 months), which further validates the antitumor efficacy of lenvatinib [39].

Drug toxicity is an indispensable factor affecting the quality of life of patients and should be taken into account when choosing optimal systemic treatment. The results in a study [40] showed that hypertension was the most common all-grade and severe AE in unresectable HCC patients who received first-line treatment with lenvatinib, which was similar to lenvatinib for other solid tumors. Interestingly, it was reported that lenvatinib-induced hypertension may be associated with better prognosis [41]. Although the specific mechanism of lenvatinib-induced hypertension is not fully understood, the possible mechanism is supposed to be associated with the interaction of neurostimulators factors, endothelin signaling pathway, renin–angiotensin aldosterone system and nitric oxide signaling pathway [42,43,44]. When compared with sorafenib, a higher incidence of decreased appetite, decreased platelet count, hypertension, hypothyroidism and proteinuria was observed in patients receiving lenvatinib, but the opposite was true for HFSR, diarrhea and rash. Overall, lenvatinib does not differ significantly from sorafenib in both the incidence of all-grade AEs and the incidence of severe AEs (grade 3–4), proving that lenvatinib is a relatively safe drug. Although lenvatinib treatment usually only lasts for months, timely recognition and management of AEs are still crucial, and avoidance of unnecessary dose reductions or interruption of the treatment is the key to ensure the antitumor efficacy.

With the increasing understanding of tumor pathogenesis, it has been found that tumor cells can evade immune system attacks through the stimulation of immune checkpoint targets; therefore, immune checkpoint inhibitors have received increasing attention in recent years [45]. In 2021, atezolizumab plus bevacizumab treatment was recommended as the first-line treatment for unresectable HCC, due to the superior survival to sorafenib [46]. However, the recently published results of a multicenter study showed that the treatment with lenvatinib is associated with a significant survival benefit compared to atezolizumab plus bevacizumab [47]. Meanwhile, the efficacy of anti-PD1 therapy in a phase III randomized trial of patients with advanced HCC has not shown statistically significant improvement due to drug resistance in some patients [48]. The mechanism of resistance was theorized in a recent study [49] arguing that the PKCa/ZFP64/CSF1 axis played a critical role in triggering immune evasion. Interestingly, lenvatinib was found to decrease PKC expression and inhibited the PKCa/ZFP64/CSF1 axis, thereby overcoming anti-PD1 resistance in HCC, while sorafenib did not [49]. Meanwhile, a real-world study reported that lenvatinib combined with PD-1-targeted immunotherapy sintilimab may lead to better long-term outcomes than lenvatinib alone [50]. Therefore, lenvatinib seems to show a survival advantage compared with sorafenib not only as monotherapy but also plays an important role in the combination of immune checkpoint inhibitors. However, the combination therapy may lead to immune checkpoint inhibitor-associated AEs, and dose personalization can reduce the related AEs and maximize patient outcomes [50,51]. Given the therapeutic advantages and synergies of lenvatinib, the potential of combination of lenvatinib and immune checkpoint inhibitors to become the new treatment trend in clinical practice is promising, which may bring hope to unresectable HCC patients.

There are several limitations in our study. First, most included studies were of a retrospective nature, which may weaken the quality of their outcomes. Second, most studies were conducted in Eastern countries, and there was a lack of studies from Western counties to perform subgroup analyses. Third, selection bias may have existed because the included articles were limited to the literature published in English. Finally, due to the limited number of studies, subgroup analysis of patients with hepatic decompensation could not be performed.

## 5. Conclusions

Our study demonstrated that lenvatinib provided better tumor response and survival benefits than sorafenib as first-line therapy for unresectable HCC patients, with a comparable incidence of AEs, presenting competitive therapeutic efficacy. More high-impact studies with larger samples are needed to further validate our conclusions.

## Figures and Tables

**Figure 1 cancers-14-05525-f001:**
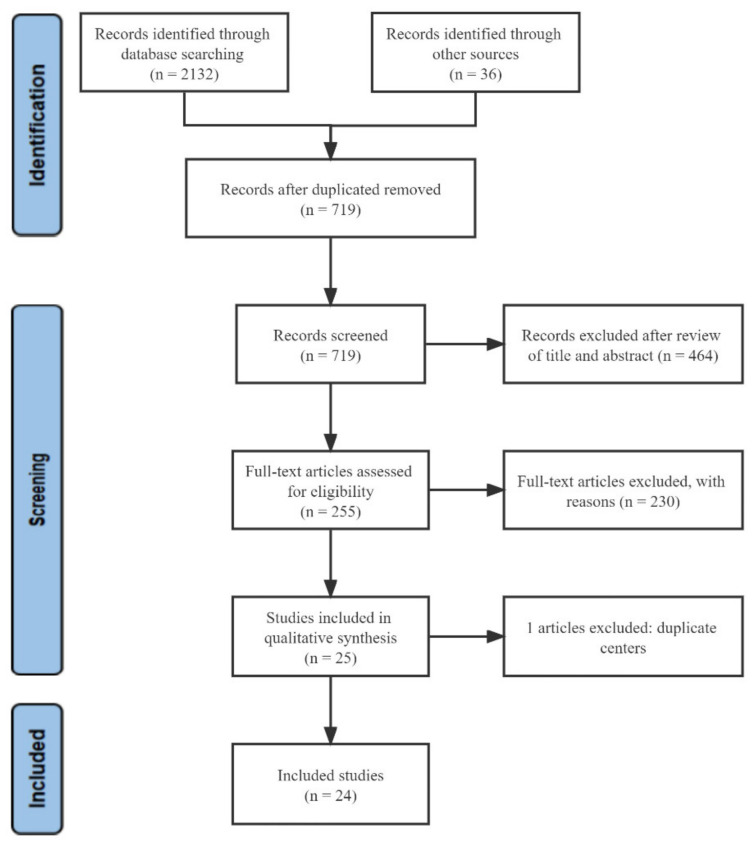
Flow diagram of the selection process.

**Figure 2 cancers-14-05525-f002:**
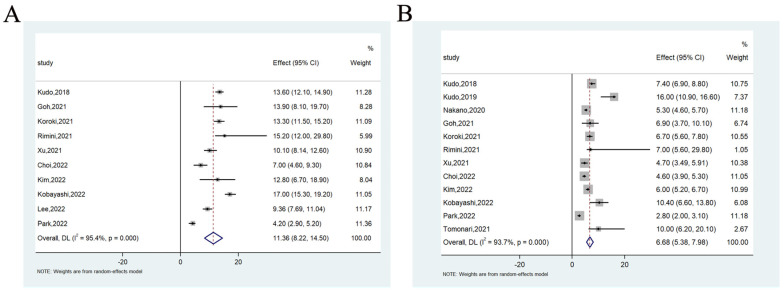
The median OS (**A**) and median PFS (**B**) of lenvatinib in the treatment of unresectable hepatocellular carcinoma.

**Figure 3 cancers-14-05525-f003:**
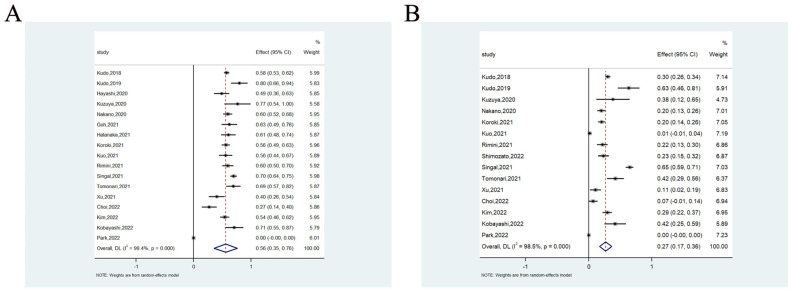
The 1-year OS rate (**A**) and 1-year PFS rate (**B**) of lenvatinib for unresectable hepatocellular carcinoma.

**Figure 4 cancers-14-05525-f004:**
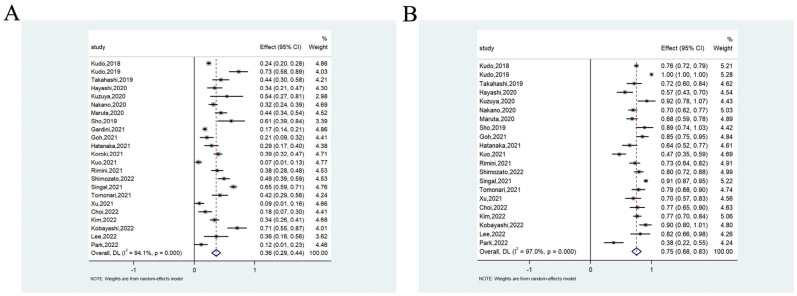
The ORR (**A**) and DCR (**B**) of lenvatinib for unresectable hepatocellular carcinoma.

**Figure 5 cancers-14-05525-f005:**
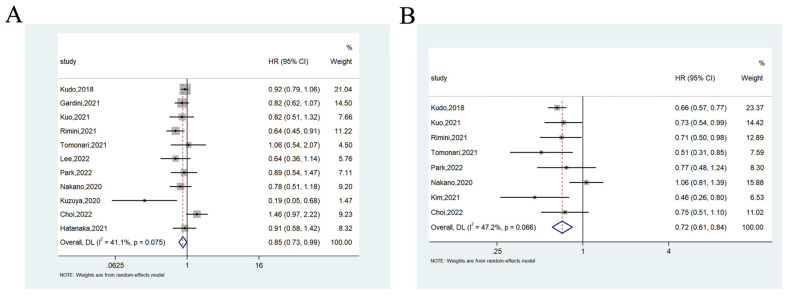
Forest plots of the OS (**A**) and PFS (**B**) associated with lenvatinib vs. sorafenib.

**Figure 6 cancers-14-05525-f006:**
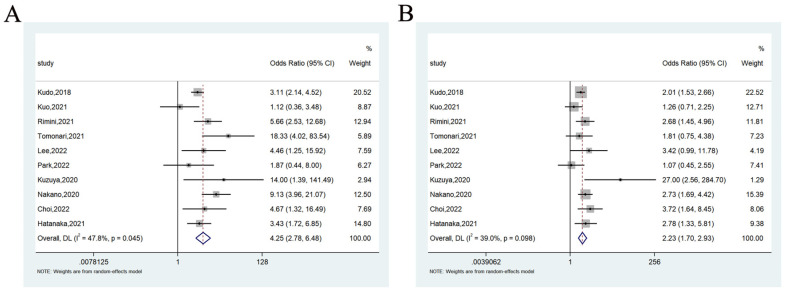
Forest plots of the ORR (**A**) and DCR (**B**) associated with lenvatinib vs. sorafenib.

**Figure 7 cancers-14-05525-f007:**
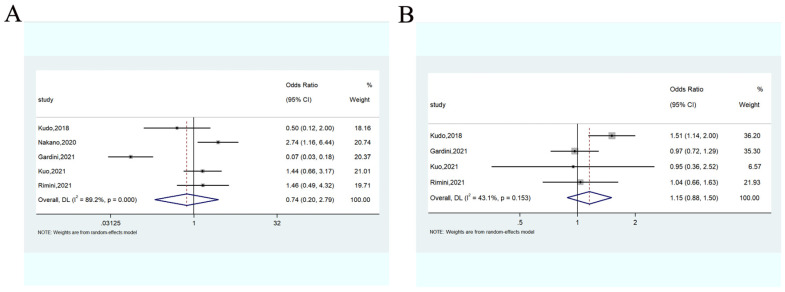
Forest plots of the all-grade (**A**) and severe adverse events (**B**) associated with lenvatinib vs. sorafenib.

**Table 1 cancers-14-05525-t001:** Baseline characteristics of patients recruited in the included studies.

Study (Years)	Country	Regimen	MFUT	Patients	Age	Female	ECOG PS: 0/1/2	Child–Pugh Class: A/B/C	HBV/HCV	MVI	EHS	Quality Score
Single-arm study												
Takahashi, 2019 [14]	Japan	Lenvatinib	6.6	50	Median = 78	12	37/12/1	47/3/0	9/18	8	13	6 ^a^
Hayashi, 2020 [15]	Japan	Lenvatinib	6.9	53	Median = 73	11	NA	53/0/0	7/15	10	18	6 ^a^
Maruta, 2020 [16]	Japan	Lenvatinib	6.7	95	47 (≤50 years)	20	90 (0 or 1)	84/11/0	13/41	25	33	7 ^a^
Sho, 2019 [17]	Japan	Lenvatinib	NA	18	Median = 75	0	11/7/0	18/0/0	3/2	NA	4	7 ^a^
Goh, 2021 [18]	Korea	Lenvatinib	7.2	48	Median = 57	7	38/NA/NA	48/0/0	NA	20	39	7 ^a^
Koroki, 2021 [19]	Japan	Lenvatinib	NA	178	94 (≤73 years)	33	169 (0 or 1)	150/NA/NA	25/70	45	64	7 ^a^
Shimozato, 2022 [20]	Japan	Lenvatinib	NA	98	Median = 76	20	91/7/0	98/0/0	19/34	14	20	6 ^a^
Singal, 2021 [21]	USA	Lenvatinib	9.1	233	Median = 62.9	75	75/147/NA	104/92/17	36/84	NA	NA	7 ^a^
Kobayashi,2022 [22]	Japan	Lenvatinib	NA	31	Median = 77	2	31/0/0	31/0/0	3/6	0	NA	6 ^a^
Double-arm study												
Kudo, 2018 [8]	Multinational	Lenvatinib vs. Sorafenib	27.7	478 476	Mean = 61.3 Mean = 61.2	73 75	304/174/0 301/175/0	475/3/0 471/5/0	251/91 228/126	329 336	291 295	2 ^b^
Kudo, 2019 [23]	Japan	Lenvatinib vs. TACE	23	30 60	Mean = 68.2 Mean = 72.4	6 18	30/0/0 60/0/0	30/0/0 60/0/0	7/12 29/10	0 0	0 0	7 ^a^
Kuzuya, 2020 [24]	Japan	Lenvatinib vs. Sorafenib	NA	13 13	Median = 70 Median = 67	2 2	12/1/0 8/5/0	13/0/0 13/0/0	2/2 2/5	NA	3 7	6 ^a^
Nakano, 2020 [25]	Japan	Lenvatinib vs. Sorafenib	7.3 10.5	146 146	Mean = 72.8 Mean = 72.8	21 25	NA	134/12/0 137/9/0	25/77 24/81	21 21	56 55	6 ^a^
Gardini, 2021 [26]	Multinational	Lenvatinib vs. Sorafenib	15.8 30.7	360 562	Mean = 66.7 Mean = 66.2	67 105	110 (1 or 2) 197 (1 or 2)	324/NA 498	93/138 164/206	196 156	146 255	7 ^a^
Kim, 2021 [27]	Korean	Lenvatinib vs. Sorafenib	NA	44 61	Median = 56 Median = 64	17 10	41 (0 or 1) 59 (0 or 1)	36 (A and B) 56 (A and B)	27/NA 45/NA	26 23	25 32	7 ^a^
Hatanaka, 2021 [28]	Japan	Lenvatinib vs. Sorafenib	9.8 10	56 375	Median = 73.5 Median = 71.0	14 75	NA	50/6/0 304/71/0	3/32 40/224	NA	NA	6 ^a^
Kuo, 2021 [29]	Taiwan (China)	Lenvatinib vs. Sorafenib	NA	70 140	Mean = 65 Mean = 65.7	20 40	NA	68/2/0 138/2/0	36/22 75/34	33 62	28 64	7 ^a^
Rimini, 2021 [30]	Multinational	Lenvatinib vs. Sorafenib	NA	92 92	23 (<65 years) 33 (<65 years)	17 11	70/22/0 65/27/0	87/5/0 85/7/0	18/38 15/41	8 10	44 44	7 ^a^
Tomonari, 2021 [31]	Japan	Lenvatinib vs. Sorafenib	9.6	52 52	Median = 70.0 Median = 71.0	16 17	38/14/0 37/15/0	52/0/0 52/0/0	15/18 10/19	11 9	10 9	7 ^a^
Xu, 2021 [32]	China	Lenvatinib vs. Toripalimab plus HAIC	NA	47 47	19 (≤50 years) 21 (≤50 years)	6 5	4/32/11 5/35/7	47/0/0 47/0/0	42/NA 43/0	NA	18 15	6 ^a^
Choi, 2022 [33]	Korea	Lenvatinib vs. Sorafenib	4.7 6.7	44 88	Median = 58.0 Median = 58.0	4 8	23/9/NA 48/29/NA	29/13/2 63/19/6	44/NA 88/NA	NA	31 59	6 ^a^
Kim, 2022 [34]	Korea	Lenvatinib vs. Atezolizumab/Bevacizumab	7.2 7.7	146 86	Median = 62 Median = 62	22 16	105/41(1 or 2) 36/50 (1 or 2)	127/19/0 82/4/0	90/19 62/3	76 43	91 37	7 ^a^
Lee, 2022 [35]	Taiwan (China)	Lenvatinib vs. Sorafenib	NA	22 44	Mean = 63.95 Mean = 63.77	4 8	NA	22/0/0 44/0/0	12/6 24/13	13 25	11 23	7 ^a^
Park, 2022 [9]	Korea	Lenvatinib vs. Sorafenib	3.7	34 60	Mean = 62 Mean = 65	5 13	NA	0/30/4 0/56/4	26/NA 43/NA	21 40	23 37	6 ^a^

Abbreviations: MFUT, median follow-up time; ECOG, Eastern Cooperative Oncology Group; PS, performance status; HBV, hepatitis B virus; HCV, hepatitis C virus; MVI, macroscopic vascular invasion; EHS, extrahepatic spread; ^a^, quality assessed using Newcastle–Ottawa Scale (NOS); ^b^, study quality assessed using JADAD score.

**Table 2 cancers-14-05525-t002:** Adverse events of lenvatinib in patients with unresectable hepatocellular carcinoma.

Adverse Events	All Grades			Grades 3 and 4		
	No. of Studies Included	Total Events	Total Patients	No. of Studies Included	Total Events	Total Patients
Diarrhea	19	493	2015	14	53	1726
Hypertension	18	604	1664	14	174	1397
PPES	18	552	1869	12	35	1220
Fatigue	17	586	1642	12	80	1353
Proteinuria	17	394	1633	13	83	1366
Decreased appetite	16	573	1568	12	76	1301
Hypothyroidism	16	343	1602	12	9	1335
Decreased PC	10	187	1030	7	36	909
Rash	10	82	1048	9	2	1030
Elevated AST	10	118	1045	9	47	995
Hepatic encephalopathy	7	33	431	5	12	363
Bilirubin elevation	7	79	476	7	14	476
Dysphonia	6	138	780	6	2	780
Nausea	6	133	812	5	5	762
Vomiting	6	99	817	6	7	817
Abdominal pain	5	113	668	5	9	668
Decreased weight	5	203	698	4	38	680
Fever	3	4	83	2	0	65
Stomatitis	3	8	101	2	0	83
Alopecia	3	14	572	3	0	572
Constipation	2	78	520	2	3	520
Increased SCR	2	4	62	2	0	62
Decreased albumin	2	4	83	1	0	65

Abbreviations: PPES, palmar–plantar erythrodysaesthesia; PC, platelet count; AST, aspartate aminotransferase; SCR, serum creatinine.

## Data Availability

All data are available in the manuscript and its supplements.

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
