# Peer review of "Lenvatinib as First-Line Treatment for Unresectable Hepatocellular Carcinoma: A Systematic Review and Meta-Analysis"

_cancers, 2022, doi:10.3390/cancers14225525_

Round 1

Reviewer 1 Report

We recommend some changes:
- We believe this article is suitable for publication in the journal although major revisions are needed. The main strengths of this paper are that it addresses an interesting and very timely question and provides a clear answer, with some limitations.
- A linguistic revision by a professional service should be performed since there are some grammar mistakes and oversights to be corrected.
- Discussion section: Very interesting and timely discussion. Of note, the authors should expand the Discussion section, including a more personal perspective to reflect on. For example, they could answer the following questions - in order to facilitate the understanding of this complex topic to readers: what potential does this study hold? What are the knowledge gaps and how do researchers tackle them? How do you see this area unfolding in the next 5 years? We think it would be extremely interesting for the readers.

- The study explores a timely topic but a major issue should be further underlined. The included study explored lenvatinib as first-line treatment. This TKI is not the current standard due to the advent of immune-based combinations such as atezolizumab plus lenvatinib. The authors should further discuss this issue.

- The background of the changing scenario of medical treatment in HCC should be better discussed, and some recent papers regarding this topic should be included (PMID: 34431725 ;  PMID: 32684988 ).
Major changes are necessary.

Author Response

Reviewer #1 (Comments to the Author (Required)): 

  1. A linguistic revision by a professional service should be performed since there are some grammar mistakes and oversights to be corrected.

Response: Thank you for your important comments. We have corrected the error in the modified manuscript and asked a professional English editing to review the article.

  1. Discussion section: Very interesting and timely discussion. Of note, the authors should expand the Discussion section, including a more personal perspective to reflect on. For example, they could answer the following questions - in order to facilitate the understanding of this complex topic to readers: what potential does this study hold? What are the knowledge gaps and how do researchers tackle them? How do you see this area unfolding in the next 5 years? We think it would be extremely interesting for the readers.

Response: Thank you for your professional comments, so that our study has been further deepened and enriched. We have discussed these topics in the modified manuscript.

  1. The study explores a timely topic but a major issue should be further underlined. The included study explored lenvatinib as first-line treatment. This TKI is not the current standard due to the advent of immune-based combinations such as atezolizumab plus lenvatinib. The authors should further discuss this issue.

Response: Thank you for your professional comments. We have discussed this interesting topic further in the modified manuscript.

  1. The background of the changing scenario of medical treatment in HCC should be better discussed, and some recent papers regarding this topic should be included (PMID: 34431725 ; PMID: 32684988 ).

Response: Thank you for your professional comments. The background of the changing scenario of medical treatment in HCC has been further enriched, and some important relevant papers has been included.

Reviewer 2 Report

Authors reviewed and conducted meta-analysis aimed to provide the most updated evidence about the efficacy and safety of lenvatinib as a first-line treatment for unresectable HCC. One phase III randomized controlled trial and 23 retrospective studies of 2438 patients were eligible for analysis. Authors reported the pooled median overall survival (OS), median progression-free survival (PFS), 1-year OS rate, 1-year PFS rate, objective response rate (ORR), and disease control rate (DCR) for patients in 27 first-line treatment with Lenvatinib. Also authors demonstrated that lenvatinib can provide better tumor responses and survival benefits than sorafenib as a first-line treatment for unresectable HCC with a comparable incidence of adverse events. Results seemed to be reasonable and meaningful as RWD analysis.

Major points

1.        Clinical outcomes was evaluated without grouping patients within or without protocol criteria. As RWD analysis, this evaluation is critically important to know how cancer therapy work in real world clinical practice within protocol criteria, and then to provide new information how cancer therapy work in real world clinical practice without protocol criteria to select therapy option against those patients.

2.        Study population varied according to clinical outcomes for each analysis. Authors should select candidate studies at the first for meta-analysis that have same sets of clinical and meta data.

Minor points

1.        There were several manuscripts published as RWD for Lenvatinib including one by Welland S. e al., [Liver Cancer 2022;11:219–232]. Those manuscripts were not appropriately cited and discussed.

2.        As to tumor response, it is not clear which methods was used to determine it (mRECIST or RECIST1.1, central review or investigator assessments)

Author Response

Reviewer #2 (Comments to the Author (Required)): 

  1. Clinical outcomes was evaluated without grouping patients within or without protocol criteria. As RWD analysis, this evaluation is critically important to know how cancer therapy work in real world clinical practice within protocol criteria, and then to provide new information how cancer therapy work in real world clinical practice without protocol criteria to select therapy option against those patients.

Response: Thank you for your professional comments. We have conducted subgroup analyses to explore the efficacy of lenvatinib in the real world according to whether patients met the REFLECT criteria.

  1. Study population varied according to clinical outcomes for each analysis. Authors should select candidate studies at the first for meta-analysis that have same sets of clinical and meta data.

Response: Thank you for your professional comments. We have conducted subgroup analyses to explore the efficacy of lenvatinib according to study setting.

  1. There were several manuscripts published as RWD for Lenvatinib including one by Welland S. e al., [Liver Cancer 2022;11:219–232]. Those manuscripts were not appropriately cited and discussed.

Response: Thank you for your professional comments. Because this study included patients who underwent surgery, we did not include it in our meta-analysis. However, the conclusion of this study is crucial to us, so we discuss it further in this paper.

  1. As to tumor response, it is not clear which methods was used to determine it (mRECIST or RECIST1.1, central review or investigator assessments)

Response: Thank you for your professional comments. We have explained it further in the result section.

Round 2

Reviewer 1 Report

acceptance

Reviewer 2 Report

Authors conducted additional subgroup analyses and some references were appropriately added, and those information will help leaders to understand this manuscript. I have no further comments.